# Gut microbiota profiles of treatment-naïve adult acute myeloid leukemia patients with neutropenic fever during intensive chemotherapy

Thanawat Rattanathammethee[1]*, Pimchanok Tuitemwong[2], Parameth Thiennimitr[3,4], Phinitphong Sarichai[3], Sarisa Na Pombejra[5], Pokpong Piriyakhuntorn[1], Sasinee Hantrakool[1], Chatree Chai-Adisaksopha[1], Ekarat Rattarittamrong[1], Adisak Tantiworawit[1], Lalita Norasetthada[1]

1 Division of Hematology, Department of Internal Medicine, Faculty of Medicine, Chiang Mai University, Chiang Mai, Thailand, 2 Department of Internal Medicine, Faculty of Medicine, Chiang Mai University, Chiang Mai, Thailand, 3 Department of Microbiology, Faculty of Medicine, Chiang Mai University, Chiang Mai, Thailand, 4 Research Center of Microbial Diversity and Sustainable Utilization, Chiang Mai University, Chiang Mai, Thailand, 5 Department of Microbiology, Faculty of Science, Chulalongkorn University, Bangkok, Thailand

* thanawat_r@outlook.com

## Abstract

The intestinal bacterial flora of febrile neutropenic patients has been found to be significantly diverse. However, there are few reports of alterations of in adult acute myeloid leukemia (AML) patients. Stool samples of each treatment-naïve AML patient were collected the day before initiation of induction chemotherapy (pretreatment), on the first date of neutropenic fever and first date of bone marrow recovery. Bacterial DNA was extracted from stool samples and bacterial 16s ribosomal RNA genes were sequenced by next-generation sequencing. Relative abundance, overall richness, Shannon's diversity index and Simpson's diversity index were calculated. No antimicrobial prophylaxis was in placed in all participants. Ten cases of AML patients (4 male and 6 female) were included with a median age of 39 years (range: 19–49) and all of patients developed febrile neutropenia. Firmicutes dominated during the period of neutropenic fever, subsequently declining after bone marrow recovery a pattern in contrast to that shown by Bacteroidetes and Proteobacteria. *Enterococcus* was more abundant in the febrile neutropenia period compared to pretreatment (mean difference +20.2; $p < 0.0001$) while *Escherichia* notably declined during the same period (mean difference -11.2; $p = 0.0064$). At the operational taxonomic unit (OTU) level, there was a significantly higher level of overall richness in the pretreatment period than in the febrile neutropenic episode (mean OTU of 203.1 vs. 131.7; $p = 0.012$). Both of the diversity indexes of Shannon and Simpson showed a significant decrease during the febrile neutropenic period. Adult AML patients with a first episode of febrile neutropenia after initial intensive chemotherapy demonstrated a significant decrease in gut microbiota diversity and the level of diversity remained constant despite recovery of bone marrow.

**Data Availability Statement:** All sequencing results are available from the Sequence Read Archive (SRA) database (accession number: BioProject PRJNA661595). All relevant data are within the paper and its Supporting Information files.

**Funding:** TR received the research grant from faculty of Medicine, Chiang Mai University, Thailand. (study code: MED-2559-03947) The funders had no role in study design, data collection and analysis, decision to publish, or preparation of the manuscript.

**Competing interests:** The authors have declared that no competing interests exist.

## Introduction

Up to half of patients with solid tumors and over 80% of those with hematologic malignancies develop a fever during chemotherapy-induced neutropenia [1]. Current recommended clinical practice includes broad-spectrum antibiotics at the onset of neutropenic fever (NF), despite most NF patients remaining negative as regards microbiological workup [2]. This practice of empiric antimicrobial attack rather than a mechanistic approach by precisely defined pathogenesis has led to serious adverse consequences, including antibiotic resistance and infection by *Clostridium difficile*.

Neutropenic fever in association with intensive chemotherapy is associated with iatrogenic damage to the gut microbiota. Intensive chemotherapy has also impaired gut barrier integrity, facilitating bacterial translocation leading to increasing risk of bloodstream infection [3]. Additionally, there is evidence of disruption of gut microbiota which could have damaging effects as the microbiota normally prevent pathogen colonization [4], provide tonic stimulation to gut barrier [5] and facilitate recovery from chemotherapy-induced injury after empirical antibiotics in NF patients [6]. While microbial cultures remain important in clinical practice, they possess several limitations including the issue that many microbes are difficult to culture under standard laboratory conditions. The selection pressure on microbial communities has been addressed by culturing methods where the particular species outgrows the others. The tests take several days to perform and hence are unable to inform clinical decision making in acute settings [7]. The metagenomics technology, including next-generation sequencing, is a novel innovation to identify microbes directly from samples without culturing which can overcome the limitations of traditional culturing methods Moreover, with the reference databases such as Greengenes [8], SILVA [9], RDP [10], or NCBI microbial genomes [11] identification of pertinent microbes is facilitated as is determination of their relative abundance.

The majority of previous studies into gut microbiota [12–17] were based on patients who received stem cell transplantation which results in a longer duration of the neutropenic period than chemotherapy. However it was found that gastrointestinal bacterial colonization was often affected during the first treatment course of acute leukemia both through mucosal barrier injuries, the use of broad spectra antibiotics and other antimicrobial agents [18].

This study aims to explore gut microbiota profiles in patients with NF during intensive chemotherapy to increase information regarding gut dysbiosis, imbalance in gut microbiota, and the timing and other specifics associated with antibiotic de-escalation. We designated the study to compare the changes in gut microbiota at pretreatment, during neutropenic fever and in the recovery phase of neutropenia. We aimed to test the hypothesis that the composition of gut microbiota may be altered in patients with acute myeloid leukemia (AML) who developed a first episode of neutropenic fever during the first cycle of intensive chemotherapy.

## Materials and methods

### Study population

We enrolled ten consecutive treatment-naïve Thai AML patients on to the study. These were all undergoing the first cycle of induction chemotherapy between July and September 2017 in the Hematology Unit of Maharaj Nakorn Chiang Mai Hospital, Chiang Mai University, Thailand. All patients were aged 18 to 65 years and their overall condition was judged to be suited to intensive treatment. The diagnosis of AML was defined as a greater than 20% presence of blasts of myeloid series in circulation and/or bone marrow examination in accordance with the WHO classification of myeloid neoplasm [19]. All patients received the standard induction chemotherapy " 7+3 regimen " (seven-days of Cytarabine 100 mg/m$^2$ intravenous continuous infusion over 24 hours combine with three-day of Idarubicin 12 mg/m$^2$ bolus intravenously).

All patients developed NF which was defined as single oral temperature of $\geq 101\,°F$ (38.3°C) or a temperature of $\geq 100.4\,°F$ (38°C) sustained over 1 hour plus an absolute neutrophil count (ANC) of $< 0.5 \times 10^9$/L [2]. Single agent Piperacillin/Tazobactam was the first empirical antibiotic given to NF patients and subsequent treatment was permitted on advice from the physician taking into account the condition of the patient, in accordance with international guideline [2]. Administration of granulocyte-colony stimulating factor (G-CSF) was not permitted in any of the participants.

Patients who had been previously treated with antibiotics within 90 days and/or probiotics and patients who received nasal tube feeding or parenteral nutrition during the study period were excluded from the study, as these factors are well known as impacting on the intestinal microbiota [20, 21]. Fluoroquinolones prophylaxis is not used in our center, but all of patients received a prophylactic dose of itraconazole (200 mg twice daily) and acyclovir (400 mg twice daily) for aspergillosis and herpes zoster reactivation, respectively.

The institutional ethical review board of Faculty of Medicine, Chiang Mai University, Thailand, approved the study (study code: MED-2559-03947). Written informed consent was obtained from all the participants before enrolment onto the study.

## Sample collection

Stool samples were collected from all 10 AML patients. Stool was collected using a standard stool kit including a sterile plastic cup with lid and a plastic bag with zip lock to seal all of specimens. Fecal samples were stored at -20°C prior to DNA extraction. Three episodes of stool sampling were indicated; pretreatment (at the day before starting of chemotherapy), the first day of febrile neutropenia and first day of bone marrow recovery. The bone marrow recovery was indicated by a surge of ANC more than $0.5 \times 10^9$/L for 24 to 48 hours apart in two consecutive times without any transfusion supported for maintenance of the appropriated level of red blood cells (more than 7 g/dl of hemoglobin [Hb] level) and platelet count (above of $10 \times 10^9$/L without bleeding symptoms) [22]. Diet was controlled in all participants in accordance with the hospital's dietary policy. All the samples were collected by individual patient (Fig 1).

## Bacterial stool DNA extraction

DNA extraction from the stool sample of AML patients was performed using QIAamp DNA Stool Mini Kit (Qiagen, Hilden, Germany) in accordance with the manufacturer's instructions. The DNA obtained was quantified using a spectrophotometer (NanoDrop Technologies, Wilmington, DE).

## Polymerase chain reaction and sequencing

The DNA samples were sent to Omics Sciences and Bioinformatics Center of Chulalongkorn University (Bangkok, Thailand) for the next generation sequencing (NGS) analysis.

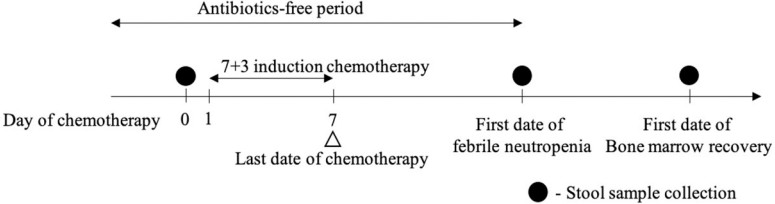

**Fig 1. Study schema.** Stool sample collection were collected from 10 consecutive adults with acute myeloid leukemia receiving 7+3 induction chemotherapy at pretreatment, first date of febrile neutropenia and first date of bone marrow recovery.

Polymerase chain reaction (PCR) amplification of the V3-V4 region of the bacterial 16s ribosomal RNA (16s rRNA) genes was performed using broad spectrum 16s rRNA primers [23] (forward primer: 5'-TCGTCGGCAGCGTCAGATGTGTATAAGAGACAGCCTACG GGNG GCWGCAG-3' and reverse primer 5'- GTCTCGTGGGCTCGGAGATGTGTATAAG AGAC AGGACTACHV GGGTATCTAATCC-3'). Amplicons were generated using a high-fidelity polymerase, 2X KAPA hot-start ready mix (KAPA Biosystems, USA). The amplification condition included an initial denaturation step 3 minutes at 94˚C, followed by 25 cycles of 98˚C for 20 seconds, 55˚C for 30 seconds, and 72˚C for 30 seconds, followed by a single final extension step at 72˚C for 5 minutes. The targeted amplicons were purified using a magnetic bead capture kit (Agencourt AMPure XP, Beckman Coulter, USA). Subsequently, the purified 16S amplicons were indexed using 2X KAPA hot-start ready mix and 5 µl of each Nextera XT index primer in a 50 µl PCR reaction, followed by 8–10 cycles of PCR condition as described above, purified using AMPure XP beads, pooled and diluted to final loading concentration at 6 pM. Sequencing was performed using the Illumina 16s MiSeq sequencing system, according to standard operating procedures, with a read length of 250 bases in paired-end sequencing mode.

## Sequencing analysis

Sequencing read quality was examined using FASTQC software [24]. Overlapping paired end reads were assembled using PEAR. FASTX-Toolkit was used to filter out assembled reads that did not have a quality score of 30 at least 90% of bases, and then reads less than 400 base-pair in length were removed. Chimeras were removed by the UCHIME method [25] as implemented in vsearch1.1.1 [26] using–uchime_ref option against chimera-free Gold RDP database. The *pick_open_reference_otus.py* command in Quantitative Insights Into Microbial Ecology (QIIME) 1.9.0 pipeline [27] was used to determine the operational taxonomic units (OTUs). These corresponded to the 16s rRNA gene sequences to address the microbial diversity and BLAST analysis was used [28] for non-redundant 16s rRNA reference sequences, which were obtained from the Ribosomal Database Project [29]. Taxonomic assignment was based on NCBI Taxonomy [30]. All of the sequencing results had been provided as an online accession number of PRJNA6611595 in Sequence Read Archive (SRA) of the National Center for Biotechnology Information (NCBI).

## Statistical analysis

All results of 16s rRNA gene sequencing were assigned to each category of bacteria as phylum to genus level. The data were entered into custom database (Excel, Microsoft Corp) and analyzed using Prism 8 software (GraphPad, Inc., La Jolla, CA). Quantitative data were reported as mean ± SD or median (range). The relative abundance, the overall richness by comparison of OTUs, and the Shannon [31] and Simpson [32] diversity indexes at phylum level were calculated. Statistical analysis included a one-way analysis of variance (alternatively, the Kruskal-Wallis test) to compare each clinical timepoint of individual patients and a paired t-test (alternatively, the Wilcoxon signed-rank test) was used to compare paired samples. Statistical corrections for multiple comparisons were performed using the original false discovery rate (FDR) method of Benjamini-Hochberg with desired false discovery rate (Q) of 0.05.

## Results

### Patients' characteristics

Ten cases of AML (4 male and 6 female) were included with a median age of 39 years (range: 19–49 years). All of patients received a 7+3 induction regimen and developed NF. Initial

empirical antibiotics was Piperacillin/Tazobactam (100%) and adjustment of antibiotics and antifungal agents was allowed in line with clinical course of individual patient by physician's decision (Fig 2). Three patients (33.3%) were defined as microbiologically significant with invasive pulmonary aspergillosis confirmed by typical radiologic finding and serum galacto-mannan (indicated as patient codes: P4, P6 and P10) and two of these were co-infected with *Pseudomonas* pneumonia (P6) and *Escherichia coli* septicemia (P10). The gastrointestinal symptoms during hospitalization included nausea (100%) and watery diarrhea (10%, P10) (Table 1). Median ANC were 2.85 x $10^9$/L (range: 1.42–7.67 x $10^9$/L), 0.04 x $10^9$/L (range: 0.01–0.43 x $10^9$/L) and 3.65 x $10^9$/L (range: 2.09–5.78 x $10^9$/L) at the day before treatment initiation, first date of febrile neutropenia and first date of bone marrow recovery, respectively. Median time to neutropenia was 11 days (range: 8–13 days) and median duration of neutropenia was 12 days (range: 7–17 days). Days of administration of antibiotics, antifungal agents and stool sample collection of individual patients are presented in Fig 2. In total, 24 stool samples were collected from 10 AML patients. The samples were assigned to three groups: (1) Pre-treatment (n = 10); (2) Febrile neutropenia (n = 9); and (3) Bone marrow recovery (n = 5). All of the missing stool samples were as a result of technical issues.

## Distribution of bacterial phyla in the gut microbiota among AML patients

Fig 3 shows the relative abundance of the bacterial phyla at each timepoint of AML treatment. Across all the samples the following five most abundant bacterial phyla were identified (Table 2): Firmicutes (41.7%), Bacteroidetes (28.7%), Proteobacteria (17.6%), Verrucomicrobia (7.8%) and Spirochaetes (2.3%). At the first date of febrile neutropenia, Firmicutes predominated rising from a median of relative abundance of 34.3% to 50.8% and subsequently declined after bone marrow recovery. In contrast, Bacteroidetes and Proteobacteria dropped at a similar level in the NF period before levelling up by the final sample collection. Levels of Verrucomicrobia and Spirochates showed very little change and there were no significant differences in relative abundance at each timepoint at phylum level.

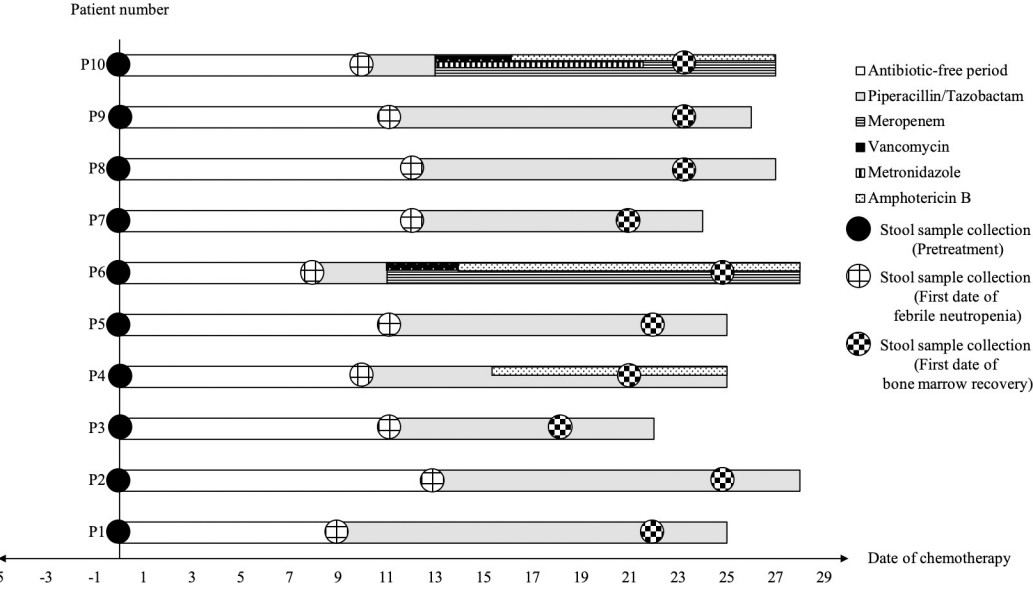

**Fig 2. Antibiotics, antifungal agents and stool sample collection of each individual patient.**

**Table 1. Characteristics of the ten AML patients included in the study.**

| Code | Sex | Age (years) | Time to neutropenia (days) | Neutropenia duration (days) | Microbiological defined events | ANC (x 10⁹/L) | | |
|---|---|---|---|---|---|---|---|---|
| | | | | | | Pre-treatment | FN | BM recovery |
| P1 | Male | 41 | 9 | 13 | none | 2.49 | 0.01 | 3.01 |
| P2 | Female | 37 | 13 | 12 | none | 3.93 | 0.03 | 5.78 |
| P3 | Male | 35 | 11 | 7 | none | 1.42 | 0.16 | 3.43 |
| P4 | Female | 24 | 10 | 11 | IPA | 7.67 | 0.08 | 5.03 |
| P5 | Male | 40 | 11 | 11 | none | 5.36 | 0.05 | 3.62 |
| P6 | Female | 19 | 8 | 17 | *Pseudomonas* pneumonia; IPA | 2.35 | 0.01 | 4.54 |
| P7 | Male | 44 | 12 | 9 | none | 1.87 | 0.02 | 3.52 |
| P8 | Female | 49 | 12 | 12 | none | 3.22 | 0.25 | 2.09 |
| P9 | Female | 44 | 11 | 13 | none | 2.38 | 0.02 | 3.68 |
| P10 | Female | 20 | 10 | 14 | *E.coli* septicemia; IPA | 3.70 | 0.43 | 3.89 |
| Median (range) | | 39 (19–49) | 11 (8–13) | 12 (7–17) | - | 2.85 (1.42–7.67) | 0.04 (0.01–0.43) | 3.65 (2.09–5.78) |

ANC, Absolute neutrophil count; FN, Febrile neutropenia; BM, Bone marrow; *E.coli*, *Escherichia coli*; IPA, Invasive pulmonary aspergillosis.

## Relative abundance at genus level

The genera within the phyla with a relative abundance over 10% (Firmicutes, Bacteroidetes and Proteobacteria) were examined to determine specific bacterial organisms (S1 Table). In the phylum Firmicutes (Fig 4A), the following bacterial genera were identified: *Enterococcus* (11.1%), *Blautia* (3.9%), *Streptococcus* (1.2%) and *Veillonella* (1.2%). *Enterococcus* was more abundant at the febrile neutropenia period (mean difference +20.2; $p < 0.0001$) and the bone marrow recovery phase (mean difference +15.4; $p = 0.0092$) compared to pretreatment. In the Bacteroidetes phylum (Fig 4B), the genus *Bacteroides* and *Parabacteroides* were extracted with a relative abundance of 21.5% and 3.8%, respectively. There was no significant change in relative abundance within the Bacteroidetes phylum across the different timepoints. In the case of Proteobacteria phylum (Fig 4C), *Sutterella*, *Escherichia* and *Klebsiella* were detected and accounted for 1.1%, 11.3% and 2.3%, respectively. Contrary to *Enterococcus* at the neutropenic fever timepoint, the *Escherichia* had significantly declined (mean difference -11.2; $p = 0.0064$)

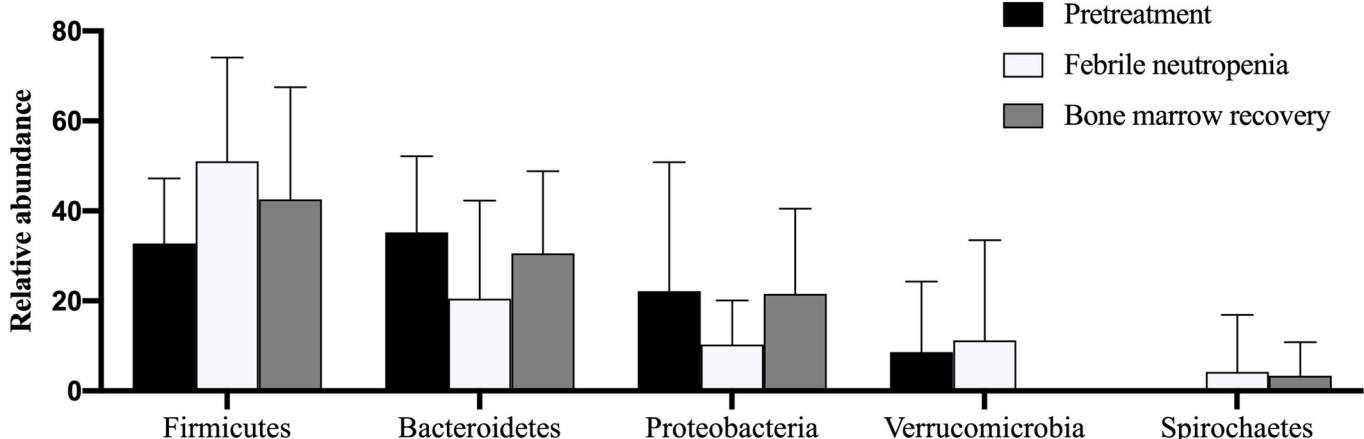

**Fig 3. Relative abundance of the bacterial phyla in gut microbiota in ten cases of acute myeloid leukemia patients.** Values shown are means ± standard deviation (SD).

**Table 2. Relative abundance of the five most abundant bacterial phyla.**

| Phylum | Relative abundance, % | Mean ± SD, % of the relative abundance | | |
|---|---|---|---|---|
| | | Pretreatment | Febrile neutropenia | Bone marrow recovery |
| Firmicutes | 41.7 | 32.7 ± 13.7 | 51 ± 21.7 | 42.5 ± 22.3 |
| Bacteroidetes | 28.7 | 35.2 ± 16 | 20.5 ± 20.5 | 30.5 ± 16.3 |
| Proteobacteria | 17.6 | 22.1 ± 27.2 | 10.3 ± 9.2 | 21.5 ± 16.9 |
| Verrucomicrobia | 7.8 | 8.6 ± 14.9 | 11.2 ± 20.9 | 0.1 ± 0.2 |
| Spirochaetes | 2.3 | 0 | 4.2 ± 11.9 | 3.3 ± 6.7 |

and remained at the lower level after the bone marrow recovery period compared to pretreatment phase (mean difference -11.8; $p$ = 0.0158).

## Richness and diversity of the gut microbiota among AML patients

To assess richness of the microbiota, the numbers of OTUs per patient were calculated (Fig 5, S2 Table). The pretreatment period showed a significantly higher mean number of OTUs compared to the febrile neutropenic episode (203.1 vs. 131.7; $p$ = 0.012). However, there were no significant differences between the OTUs at bone marrow recovery and pretreatment and first date of febrile neutropenia.

The Shannon and Simpson diversity indices were used for comparison at the phylum level. (Fig 6A and 6B). It was found that both the Shannon and Simpson diversity index indicated a significant reduction in bacterial abundance at the febrile neutropenic period in comparison to the pretreatment samples. (median of Shannon's index of 1.077 vs. 1.002; $p$ = 0.044, and median of Simpson's index of 0.628 vs. 0.521; $p$ = 0.027).

## Discussion

This study demonstrates the changes in gut microbiota in newly diagnosed adult AML patients who underwent a first cycle of induction chemotherapy with carefully controlled factors. Research indicated that the composition of the gut microbiota would be affected in all participants by the first episode of NF. We hypothesized that even the first cycle of intensive chemotherapy, which would contribute the NF, without antibiotic prophylaxis might reveal evidence of changes in intestinal bacteria flora as all of febrile neutropenia patients require empirical treatment with broad spectrum antibiotics. The gut barrier is a vulnerable site of injury in patients who have received highly intensive chemotherapy and resulting in microbiota pattern disruption. The antibiotic therapy may eradicate particular taxa which affect host immunity and the chemotherapy can extensively damage the gut barrier, the neutropenia itself also potentially affecting the intestinal bacteria of the patients [7].

The majority of previous reports regarding gut microbiota reported that a predominance of patients who received stem cell transplantation in the neutropenic period might suffer intense and longer duration of bone marrow recovery compared to other kinds of chemotherapeutic regimens. In the case of allogeneic stem cell transplant recipients, the increase of relative abundance of *Enterococcus* and *Proteobacteria* during peri-transplant period was found to be significantly associated to bloodstream infection (BSI) and likelihood of bacterial translocation [12, 33]. It was also found to be associated with a loss of diversity with domination by a single taxa with the addition issue that exposure to particular anti-anaerobic antibiotics subsequently escalated the risk of graft-versus-host disease (GVHD) and mortality rate [15, 16, 34]. In another study, pretreatment gut microbiota was used to predict chemotherapy-related blood stream infection and machine learning was used to create a BSI risk index scoring system for

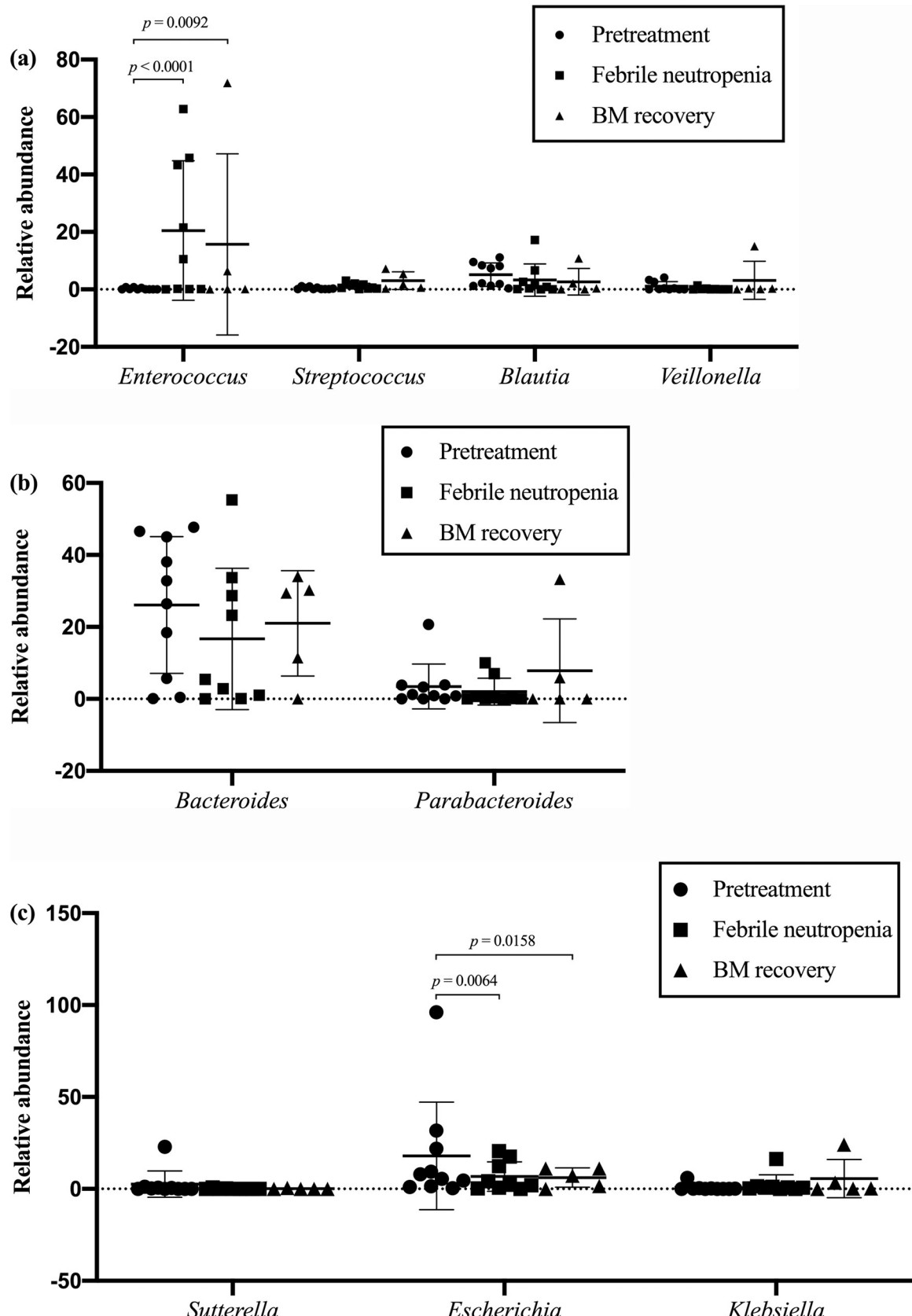

**Fig 4. Relative abundance at genus level of phylum with a relative abundance over 10%.** (a) Firmicutues; (b) Bacteroidetes; (c) Proteobacteria. Each symbol represents one individual sample. Value shown are mean ± SD and regarding *p*-value determined the significant difference after statistical corrections using the original false discovery rate (FDR) method of Benjamini-Hochberg for multiple comparisons.

non-Hodgkin lymphoma patients who underwent autologous stem cell transplantation [35]. All of these results confirmed that intestinal tract microbial diversity plays a major role in the outcomes of treatment and disruption can lead to multiple complications.

This study found a significant loss of fecal microbial diversity during the neutropenic period with domination by the phylum Firmicutes and a significant increase of *Enterococcus* at genus level. This finding was in agreement with previous stool microbiota studies in adult AML patients [6, 36, 37]. The increase of bacterial abundance in the Enterococcaceae and Streptococcaceae families in the Firmicutes phylum was previously reported as a strong predictor of infectious complications in pediatric acute lymphoblastic leukemia (ALL) and adult AML patients [38, 39].

A single large study of gut microbiota in AML patients during induction chemotherapy [40] reported that a longitudinal analysis of oral and stool microbiota measurement that could assist in the mitigation of infectious complications. A significant decrease in both oral and stool microbial diversity were observed over the course of induction chemotherapy with a strong correlation between both sites of sample collection. The patients who had a decrease in microbial diversity were significantly more likely to have a microbiologically documented infection within 90 days post treatment. A comparative study of gut dysbiosis in patients undergoing intensive chemotherapy and allogeneic hematopoietic stem cell transplantation also supported the findings of a similar loss of microbial diversity and domination of low-diversity communities by *Enterococcus* during a period of intense neutropenia [6]. Additionally, this study also noted a significant reduction in *Bacteroides* and *Escherichia* (subsets of Bacteroidetes and Proteobacteria families, respectively) in the NF period. However, there were inconclusive reports regarding the changes in the Bacteroidetes family with a limited correlation in clinical data. There is relatively little information on changing patterns in the

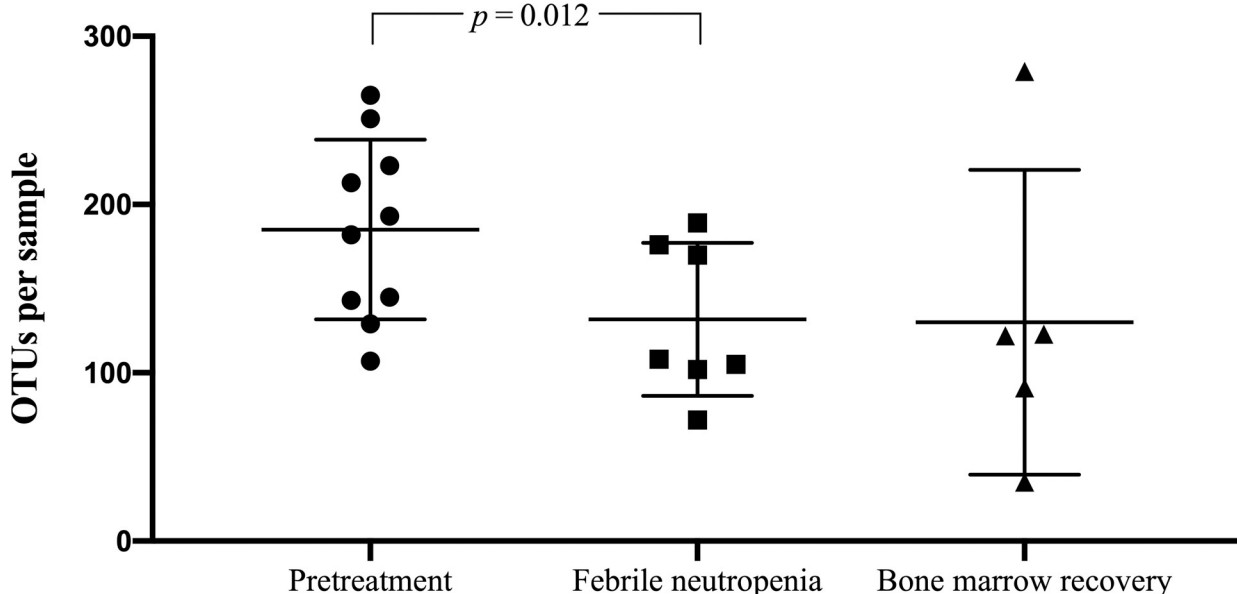

**Fig 5. Richness of the gut microbiota in acute myeloid leukemia patients.** Bacterial DNA was extracted and the 16s rRNA genes were sequenced and assigned to operational taxonomic units (OTUs). Each symbol represents one individual sample. Values shown are mean ± SD.

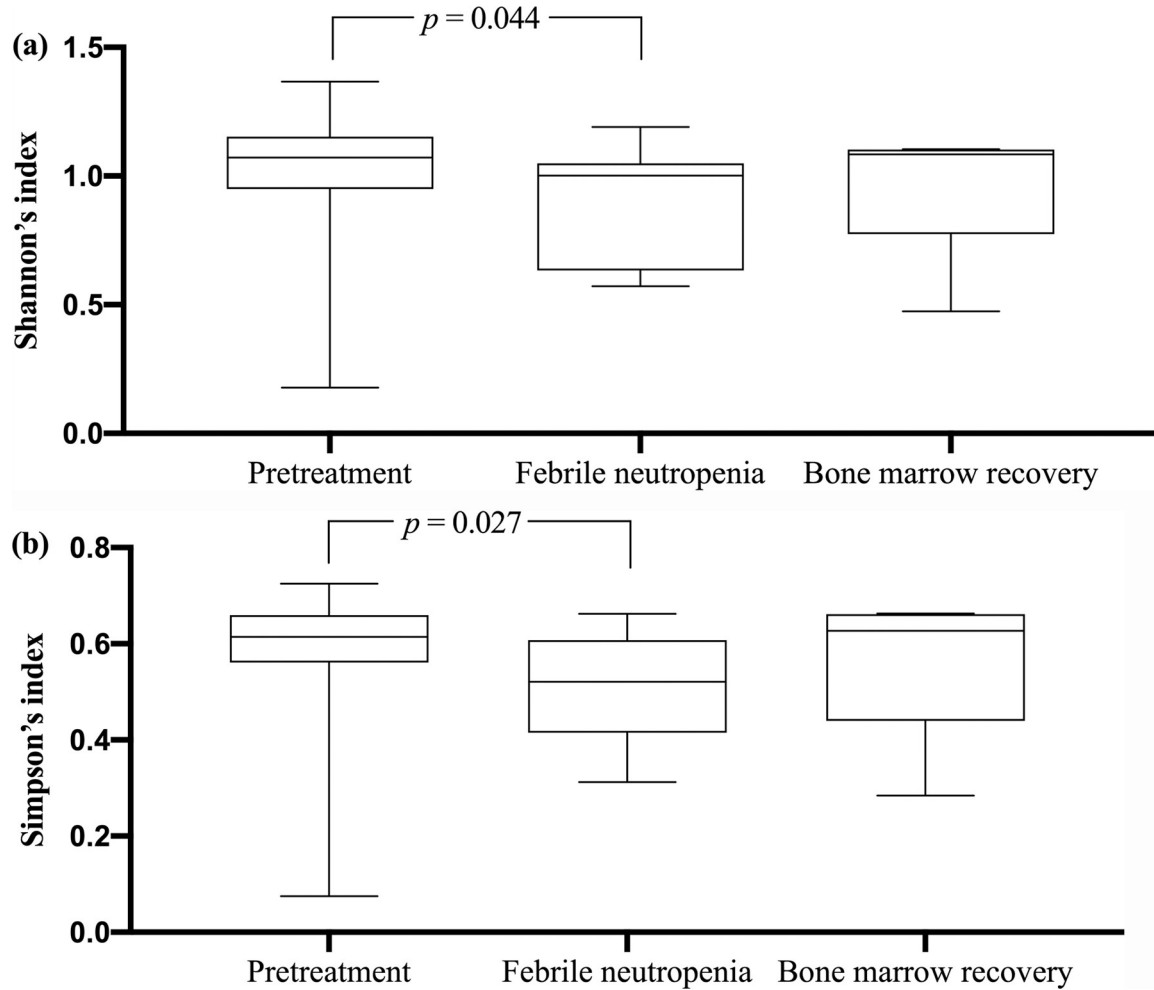

**Fig 6. Diversity indices at phylum level.** (a) Shannon's index values; (b) Simpson's index values. Comparison between groups by one-way ANOVA (comparisons between all groups) and Wilcoxon's signed-rank test (comparisons between the paired samples). The 25th and 75th percentile are shown in the box plot. The median is indicated by horizontal solid lines. The bars indicate the minimum and maximum values.

Proteobacteria family with a single report stating there was a predictive risk of febrile neutropenia. All of these studies included solely pediatric ALL patients [37, 39, 41].

Ethnicity and geographical location are also considered to influence the composition of the gut microbiota. Interindividual differences in the intestinal microbiota profiles have been predicted accurately by the location of the host individual [42] and Even in the cases where the environment is the same but the ethnicity varied there are significant differences in gut microbiota patterns [43]. Unfortunately, the gut microbiota studies carried out to investigate the association of ethnicity and geographical location were only carried out in healthy subjects therefore do not reflect the setting of patients with hematologic cancer. Also, the current knowledge of how the composition of the gut microbiota relates to patient health is principally based on investigations in European and North American populations. This may limit the generalizable properties of microbiome-based applications for personalized medicine [44]. Our study exclusively recruited patients of Asian ethnicity with a uniform pattern of a clinical course, specifically treatment-naïve patients undergoing a homogenous course of chemotherapy with no previous use of antimicrobial prophylaxis.

Our study has several limitations to consider. Firstly, the relatively small sample size of the study and missing data limits the interpretation of the results due to lack of statistical power. Despite the small sample size, the results from the study and the research warrant the carrying out of a larger study, possibly across several centers.

In conclusion, the loss of fecal microbiota diversity can be used to predict and therefore address future potential complications associated with the treatment of these vulnerable AML patients. These findings warrant the conduct of further research in this field in adult AML patients which will add valuable clinical decision making information for individualized treatment of patients as regards antimicrobial de-escalation and prediction of anticipated complications.

## Supporting information

**S1 Table. Relative abundance at genus level of phyla with a relative abundance over 10%.** Statistical corrections for multiple comparisons were performed using the original false discovery rate (FDR) method of Benjamini-Hochberg with desired false discovery rate (Q) of 0.05. (DOCX)

**S2 Table. The numbers of OTUs per sample.** (DOCX)

## Acknowledgments

We would like to thank the Omics Sciences and Bioinformatics Center of Chulalongkorn University (Bangkok, Thailand) for contributing the facility to carry out the DNA sequencing and collection of the gut microbiota data.

## Author Contributions

**Conceptualization:** Thanawat Rattanathammethee, Pimchanok Tuitemwong, Parameth Thiennimitr.

**Data curation:** Thanawat Rattanathammethee, Pimchanok Tuitemwong, Parameth Thiennimitr.

**Formal analysis:** Thanawat Rattanathammethee, Pimchanok Tuitemwong, Phinitphong Sarichai, Sarisa Na Pombejra.

**Funding acquisition:** Thanawat Rattanathammethee, Pimchanok Tuitemwong.

**Investigation:** Thanawat Rattanathammethee, Pimchanok Tuitemwong.

**Methodology:** Thanawat Rattanathammethee, Pimchanok Tuitemwong, Parameth Thiennimitr, Phinitphong Sarichai, Sarisa Na Pombejra.

**Writing – original draft:** Thanawat Rattanathammethee, Pimchanok Tuitemwong.

**Writing – review & editing:** Parameth Thiennimitr, Pokpong Piriyakhuntorn, Sasinee Hantrakool, Chatree Chai-Adisaksopha, Ekarat Rattarittamrong, Adisak Tantiworawit, Lalita Norasetthada.

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
