## [Decision Letter · Decision Letter 0]

20 Aug 2020

PONE-D-20-20999

Gut microbiota profiles of treatment-naïve adult acute myeloid leukemia patients with neutropenic fever during intensive chemotherapy

PLOS ONE

Dear Dr. Rattanathammethee,

Thank you for submitting your manuscript to PLOS ONE. After careful consideration, we feel that it has merit but does not fully meet PLOS ONE’s publication criteria as it currently stands. Therefore, we invite you to submit a revised version of the manuscript that addresses the points raised during the review process.

The reviewers and I feel that the manuscript, as currently presented, suffers from a lack of clinical connection with the microbiome data. Further, revision of statistical approaches and presentation of the data are suggested. Due to the small sample size and primarily descriptive findings, revision to a shorter format is also encouraged.

We look forward to receiving your revised manuscript.

Kind regards,

Christopher Staley, Ph.D.

Academic Editor

PLOS ONE

Journal Requirements:

2.We note that you are reporting an analysis of a microarray, next-generation sequencing, or deep sequencing data set. PLOS requires that authors comply with field-specific standards for preparation, recording, and deposition of data in repositories appropriate to their field. Please upload these data to a stable, public repository (such as ArrayExpress, Gene Expression Omnibus (GEO), DNA Data Bank of Japan (DDBJ), NCBI GenBank, NCBI Sequence Read Archive, or EMBL Nucleotide Sequence Database (ENA)). In your revised cover letter, please provide the relevant accession numbers that may be used to access these data. For a full list of recommended repositories, see http://journals.plos.org/plosone/s/data-availability#loc-omics or http://journals.plos.org/plosone/s/data-availability#loc-sequencing.

Reviewers' comments:

Reviewer's Responses to Questions

**Comments to the Author**

1. Is the manuscript technically sound, and do the data support the conclusions?

Reviewer #1: Partly

Reviewer #2: Yes

2. Has the statistical analysis been performed appropriately and rigorously? 

Reviewer #1: No

Reviewer #2: No

3. Have the authors made all data underlying the findings in their manuscript fully available?

Reviewer #1: Yes

Reviewer #2: No

4. Is the manuscript presented in an intelligible fashion and written in standard English?

Reviewer #1: No

Reviewer #2: No

5. Review Comments to the Author

Reviewer #1: The authors present a report in which they follow the microbiological flora ny NGS sequencing of bacterial DNA/rRNA in the guts of patients with AML . They compare finds pretreatment , the time of neutropenic fever (F/N) as well as upon recovery. They make some observations concerning the changes noted; in particular, they seem to show that bacterial diversity declines between pre-treatment and F/N

Comments

1. The abstract and the entire paper suffer from severe problems with grammar and syntax; these need to be addressed, most likely by a native English speaker.

2. Abstract. This needs to be sufficiently re-worked to give a better idea to the reader of the quantitative findings. Further no mention is made of antibiotic usage, particularly prophylaxis, which could obviously affect the results.

3. The Introduction should state what advantage the sequencing technology has over culturing methods to determine gut bacterial diversity.

4. Is the fact that all patients had invasive pulmonary aspergillosis unusual ( or so that is stated in the text, but not related in the table)?

5. What does 3 patients … microbiologically defined mean?

6. The pretreatment neutrophil count seemed remarkably homogeneous for a group of patients with AML. The AML characteristic should be provided.

7. Table 2 what do the numbers in the leftmost column represent?

8. In the discussion section the authors really don't give a good explanation of why the flora of the gut changed despite the lack of use of antibacterial prophylaxis (and before any specific therapy for fever neutropenia was given)

Reviewer #2: The authors present a study examining the bacterial composition of stool samples collected from patients undergoing initial induction chemotherapy treatment for AML. Three time points were analyzed from 10 patients, with 100% and 90% of samples successfully analyzed at baseline and at onset of febrile neutropenia, and 50% of samples analyzed at bone marrow recovery. The methodologies reported are appropriate and the manuscript is overall well-written. I have the following comments:

- The study overall is fairly limited. The sample size is only 10 patients, with significant drop-off of samples collected at the 3rd time point. Such a small sample size doesn't allow for exploration of microbiome associations with different clinical outcomes. Larger retrospective cohorts have already been published, and it is a but unclear what this current study adds to the field.

- Comparisons of microbiome parameters (abundances of taxa or diversity) should be adjusted for multiple comparisons, using for example the Benjamini-Hochberg method. They currently don't appear to be adjusted.

- Conveying bacterial abundances using tables is probably not as effective as graphically. Consider preparing stacked bar graphs for individual samples.

- Data underlying the findings described are not fully available in the manuscript, supplementary materials, or in an online repository.

- Consider asking a native English speaker to edit the manuscript.

6. PLOS authors have the option to publish the peer review history of their article (what does this mean?). If published, this will include your full peer review and any attached files.

Reviewer #1: No

Reviewer #2: No

---

## [Author Response · Author response to Decision Letter 0]

5 Sep 2020

Response to reviewers

=========

Reviewer #1: The authors present a report in which they follow the microbiological flora ny NGS sequencing of bacterial DNA/rRNA in the guts of patients with AML . They compare finds pretreatment , the time of neutropenic fever (F/N) as well as upon recovery. They make some observations concerning the changes noted; in particular, they seem to show that bacterial diversity declines between pre-treatment and F/N

Comments

1. The abstract and the entire paper suffer from severe problems with grammar and syntax; these need to be addressed, most likely by a native English speaker.

Response: The manuscript has already been revised by English language team.

2. Abstract. This needs to be sufficiently re-worked to give a better idea to the reader of the quantitative findings. Further no mention is made of antibiotic usage, particularly prophylaxis, which could obviously affect the results.

Response: Addition of quantitative findings and usage of antibiotics have been addressed.

3. The Introduction should state what advantage the sequencing technology has over culturing methods to determine gut bacterial diversity.

Response: The advantage of sequencing technology over traditional culturing methods has been addressed.

4. Is the fact that all patients had invasive pulmonary aspergillosis unusual ( or so that is stated in the text, but not related in the table)?

Response: Only 3 patients had invasive pulmonary aspergillosis. I have rewritten this statement into " Three patients (33.3%) were defined as microbiologically significant with invasive pulmonary aspergillosis confirmed by typical radiologic finding and serum galactomannan (indicated as patient's code: P4, P6 and P10)".

5. What does 3 patients … microbiologically defined mean?

Response: I have rewritten this statement into "Three patients (33.3%) were defined as microbiologically significant with invasive pulmonary aspergillosis confirmed by typical radiologic finding and serum galactomannan (indicated as patient's code: P4, P6 and P10)".

6. The pretreatment neutrophil count seemed remarkably homogeneous for a group of patients with AML. The AML characteristic should be provided.

Response: I have provided the AML patients' characteristic in Table 1. The median of pretreatment absolute neutrophil count was 2.85 x 109 /L (1.42-7.67) with the range of 1.42 x 109 /L (minimum) to 7.67 x 109 /L (maximum).

7. Table 2 what do the numbers in the leftmost column represent?

Response: The leftmost column in table 2 represent the name of each bacterial phyla and next column is the percentage of microbial relative abundance among five most abundant stool bacterial phyla.

8. In the discussion section the authors really don't give a good explanation of why the flora of the gut changed despite the lack of use of antibacterial prophylaxis (and before any specific therapy for fever neutropenia was given)

Response: I have explained more detailed on the changes of gut microbiota in the discussion section. (We hypothesized that even the first cycle of intensive chemotherapy, which would contribute to the NF, and without antibiotic prophylaxis might reveal evidence of changes in intestinal bacteria flora as all of febrile neutropenia patients require empirical treatment with broad spectrum antibiotics. Gut barrier is a vulnerable site of injury after receiving highly intensive chemotherapy and resulting microbiota pattern disruption. The antibiotic therapy may eradicate particular taxa which affect host immunity and the chemotherapy can extensively damage the gut barrier, the neutropenia itself also potentially affecting the intestinal bacteria of the patients.)

=========

Reviewer #2: The authors present a study examining the bacterial composition of stool samples collected from patients undergoing initial induction chemotherapy treatment for AML. Three time points were analyzed from 10 patients, with 100% and 90% of samples successfully analyzed at baseline and at onset of febrile neutropenia, and 50% of samples analyzed at bone marrow recovery. The methodologies reported are appropriate and the manuscript is overall well-written. I have the following comments:

- The study overall is fairly limited. The sample size is only 10 patients, with significant drop-off of samples collected at the 3rd time point. Such a small sample size doesn't allow for exploration of microbiome associations with different clinical outcomes. Larger retrospective cohorts have already been published, and it is a but unclear what this current study adds to the field.

Response: This study conducted solely treatment-naïve AML patients who homogenously experienced the first episode of neutropenic fever and currently not received antimicrobial prophylaxis. Unfortunately, the limitation of clinical correlation cannot be eliminated but to point out the microbial composition changes even in the first dose of AML treatment which microbiota diversity remained constant despite recovery of bone marrow. Moreover, the study also expand the dataset of gut microbiota profiles in adult Asian ethnicity with AML.

- Comparisons of microbiome parameters (abundances of taxa or diversity) should be adjusted for multiple comparisons, using for example the Benjamini-Hochberg method. They currently don't appear to be adjusted.

Response: In terms of correction of multiple comparisons, Benjamini-Hochberg method has been introduced to the manuscript. 

- Conveying bacterial abundances using tables is probably not as effective as graphically. Consider preparing stacked bar graphs for individual samples.

Response: I have converted table 3 to the stacked bar graph for individual sample and this table has been moved to supplementary data (Table S1). 

- Data underlying the findings described are not fully available in the manuscript, supplementary materials, or in an online repository.

Response: All of the sequencing results had been provided as an online accession of Sequence Read Archrive (SRA), as BioProject number PRJNA661595 (https://www.ncbi.nlm.nih.gov/Traces/study/?acc=PRJNA661595),

and supplementary material has been added.

- Consider asking a native English speaker to edit the manuscript.

Response: The manuscript has already been revised by English language team.

---

## [Decision Letter · Decision Letter 1]

14 Oct 2020

Gut microbiota profiles of treatment-naïve adult acute myeloid leukemia patients with neutropenic fever during intensive chemotherapy

PONE-D-20-20999R1

Dear Dr. Rattanathammethee,

We’re pleased to inform you that your manuscript has been judged scientifically suitable for publication and will be formally accepted for publication once it meets all outstanding technical requirements.

Kind regards,

Christopher Staley, Ph.D.

Academic Editor

PLOS ONE

Additional Editor Comments (optional):

Reviewers' comments:

Reviewer's Responses to Questions

**Comments to the Author**

1. If the authors have adequately addressed your comments raised in a previous round of review and you feel that this manuscript is now acceptable for publication, you may indicate that here to bypass the “Comments to the Author” section, enter your conflict of interest statement in the “Confidential to Editor” section, and submit your "Accept" recommendation.

Reviewer #2: All comments have been addressed

2. Is the manuscript technically sound, and do the data support the conclusions?

Reviewer #2: Yes

3. Has the statistical analysis been performed appropriately and rigorously? 

Reviewer #2: Yes

4. Have the authors made all data underlying the findings in their manuscript fully available?

Reviewer #2: Yes

5. Is the manuscript presented in an intelligible fashion and written in standard English?

Reviewer #2: Yes

6. Review Comments to the Author

Reviewer #2: The authors have responded to the comments and the manuscript is improved. I have no further concerns.

7. PLOS authors have the option to publish the peer review history of their article (what does this mean?). If published, this will include your full peer review and any attached files.

Reviewer #2: No

---

## [Editor Report · Acceptance letter]

19 Oct 2020

PONE-D-20-20999R1 

Gut microbiota profiles of treatment-naïve adult acute myeloid leukemia patientswith neutropenic fever during intensive chemotherapy 

Dear Dr. Rattanathammethee:

I'm pleased to inform you that your manuscript has been deemed suitable for publication in PLOS ONE. Congratulations! Your manuscript is now with our production department. 

Kind regards, 

on behalf of

Dr. Christopher Staley 

Academic Editor

PLOS ONE